# “Not Alone in Loneliness”: A Qualitative Evaluation of a Program Promoting Social Capital among Lonely Older People in Primary Health Care

**DOI:** 10.3390/ijerph18115580

**Published:** 2021-05-23

**Authors:** Laura Coll-Planas, Dolors Rodríguez-Arjona, Mariona Pons-Vigués, Fredrica Nyqvist, Teresa Puig, Rosa Monteserín

**Affiliations:** 1Fundació Salut i Envelliment (Foundation on Health and Ageing), Universitat Autònoma de Barcelona, 08041 Barcelona, Spain; lrodriguez79@hotmail.com; 2Institute of Biomedical Research (IIB Sant Pau), 08041 Barcelona, Spain; tpuig@santpau.cat (T.P.); rmonteserin@eapsardenya.cat (R.M.); 3Servei Català de la Salut (CatSalut), Planning and Assessment Area, 08028 Barcelona, Spain; mariona.pons@catsalut.cat; 4Nursing Department at the Faculty of Nursing, Universitat de Girona, 17003 Girona, Spain; 5Faculty of Education and Welfare Studies, Social Policy, Åbo Akademi University, 65101 Vaasa, Finland; fredrica.nyqvist@abo.fi; 6Universitat Autònoma de Barcelona, 08193 Bellaterra (Cerdanyola del Vallès), Spain; 7Epidemiology and Public Health Department, Hospital de la Santa Creu i Sant Pau, 08041 Barcelona, Spain; 8Equip d’Atenció Primària Sardenya, EAP Sardenya, 08025 Barcelona, Spain

**Keywords:** ageing, qualitative research, primary health care, loneliness, social capital

## Abstract

The weekly group-based program “Paths: from loneliness to participation” was conducted face-to-face over 15 sessions by nurses, social workers and volunteers in primary care in Catalonia (Spain) to alleviate loneliness among older people by promoting peer support and participation in community assets. We aimed at exploring participants’ experiences of loneliness and participation prior to the program and its perceived benefits. The qualitative design was descriptive-interpretative. Data were collected through three focus groups and 41 interviews applying a semistructured topic guide involving 26 older participants, six professionals and nine volunteers. Participant-observation of all sessions involved the 38 older people who started the program. A thematic content analysis was applied. Older persons with diverse profiles of loneliness and participation explained different degrees of decrease in loneliness, an increase in participation in local community assets, companionship, peer support and friendship, and an empowerment process. Successful cases reported improvements in mental wellbeing and recovering the sense that life was worth living. Loneliness persisted among some widowed participants and vulnerabilities hampered some benefits. Participants, professionals and volunteers reported different degrees of success in older people to alleviate loneliness by enhancing social relationships and activities through complex processes interrelated with health and socioeconomic factors.

## 1. Introduction

Loneliness is defined as a negative feeling due to the perception that the social needs of the person are not corresponded, neither in quantity nor in quality, by the social relationships that the person has [1]. In the last years, the public awareness and the scientific concern about the phenomenon of loneliness has increased [2]. Furthermore, the current SARS-COV2 pandemic has accentuated the value of social interactions and social support and the need to alleviate loneliness among older people [3]. Older people undergo major changes in their social environment mainly due to retirement, widowhood, loss of peers, and age-related disability [4]. Likewise, three ageing crises are related to loneliness: the identity (no longer feeling like who they used to be), autonomy (not being able to do what they used to do) and belonging crises (not belonging to the places and groups of persons to which they used to belong) [5].

At a personal level, several risk factors related with sociodemographic characteristics and health status are associated with loneliness: being female, living alone, limited education, small social network, low self-efficacy, poor self-rated health, depression and recent bereavement (often due to widowhood) [6,7]. Geographically, loneliness differs across Europe being higher in Southern countries such as Spain [8]. In Southern Europe, the cultural emphasis on family and social relationships generates high expectations and social needs that might be more challenging to fulfil. Moreover, active participation in social organizations is seen as vital to build relationships while ageing, but it is lower among older people in Spain in a European comparative perspective [9,10,11,12]. From a policy perspective, the WHO Active Ageing and Healthy Ageing paradigms have encouraged for the last 20 years to enhance social participation and social networks for ageing people [13,14]. In this vein, Putnam’s definition of social capital has been adapted to older age placing more relevance on the interaction between individuals at the micro level. Accordingly, social capital is an umbrella concept that involves individual (family and friends) and collective social resources (e.g., neighborhoods), their structural (e.g., social networks, social contacts and participation) and cognitive aspects (e.g., social support and sense of belonging) [15,16,17]. However, the processes involved in the promotion of social capital in ageing, including social relationships and participation, remain unclear [4].

Certain intervention characteristics are related to a higher efficiency at reducing loneliness, such as theory-driven interventions [18,19,20]. However, it is not yet clear which theory supports more effective interventions. The loneliness model supports cognitive behavioral therapy to correct deficits in social skills and address maladaptive social cognition [21]. On the contrary, the empowerment theory considers that loneliness is potentially alleviated through empowering lonely older people to increase their self-esteem and feeling of mastery over their own life [22,23,24]. Moreover, theories of behavior change might be used to better understand how to promote social relations and social participation [25,26]. Finally, the most widely applied strategy among older people to tackle loneliness is increasing social support. However, controlled trials evaluating this intervention strategy are scarce [21].

Regarding intervention effectiveness, a systematic review on interventions based on social capital targeting older people showed few and diverse trials assessing the impact on loneliness and they were generally ineffective [27]. However, some successful studies targeted complex cases of loneliness, and social capital interventions successfully increased quality of life, well-being and self-perceived health among lonely older people. In this vein, the intervention Circle of Friends in Finland, which focused on empowering lonely older people, achieved successful improvements in a wide range of health outcomes including mortality, but not in loneliness [24]. Their qualitative analysis showed how lonely participants built trust and encouragement and continued to meet [28]. A program based on facilitating community knowledge and networking among older migrants in Japan through volunteers as gatekeepers, decreased loneliness and increased social support [29].

In Spain, the program “Paths: from loneliness to participation” was designed, conducted and evaluated to alleviate loneliness among older people attending primary health care [30]. The intervention promoted peer support and social participation by enhancing engagement in activities in community assets. The intervention was evaluated with mixed methods. According to the quantitative evaluation, loneliness decreased and social participation and support significantly increased [30].

In summary, despite a diversity of programs in place around the world and isolated successful results, evidence and detailed understanding on whether and how programs decrease loneliness is lacking, as well as how the characteristics of the target population influence the impact. Likewise, while previous literature clearly suggests health beneficial effects of social capital, less is known about social capital interventions and how social capital can be built for health promoting purposes. Qualitative evaluation of interventions to explore participants and professionals’ perspective on the experiences and perceived benefits is a complementary approach to the quantitative evaluation of objective impacts that can help understand the processes and interpret the effects of the programs.

Therefore, this paper reports the qualitative evaluation of the program “Paths: from loneliness to participation” aimed at exploring participants’ experiences of loneliness and social participation prior to the program; and describing its perceived benefits on loneliness, social participation, and support and health according to participants’ experience, volunteers and professionals’ observations.

## 2. Materials and Methods

The study was conducted adhering to the rigor and quality criteria for qualitative research: description of context, of participants and of the research process, methodological adequacy, triangulation of data and reflexivity of the research team [31]. Moreover, it is reported according to the Standards for Reporting Qualitative Research [32].

Throughout the paper, “participants” refers to older people participating in the program and “informants” comprises all agents involved: participants, volunteers, and professionals.

### 2.1. Design

Following a constructivist research paradigm, a qualitative study with a phenomenological approach was chosen, in order to explore the lived experiences of the involved agents applying a descriptive-interpretative design.

The perceived benefits of the program were identified among participants according to their experiences and then triangulated with the perceptions of volunteers and health and social care professionals and with the researchers’ observations.

This research applies the social capital theory adapted to ageing by Nyqvist and Forsman [33].

### 2.2. Description of the Program

The “Paths: from loneliness to participation” program is theory-driven and was designed around the mentioned operationalization of the social capital theory applied to ageing with the goal to alleviate loneliness among older people by promoting peer support and participation in local community assets [15,30,34]. It was conducted from December 2011 to July 2012 in primary health and social care centers in Catalonia (Spain). Sessions were one and half hours long and took place once a week during 15 weeks. The program has been previously described in detail, as well as its overall intervention framework [30]. In summary, older people with low or no participation in social activities and suffering from loneliness at least sometimes were referred by primary health and social care professionals to the group. The group met face-to-face and was led by social workers or nurses from the primary health or social care center. The group dynamic was grounded on active participation in line with the empowerment theory. Along the 15 sessions, peer support was promoted through sharing opinions and experiences around loneliness and participation prompted by a diversity of pictures. Furthermore, older people active in the same neighborhood were involved as volunteers to connect participants with the local community assets. As a group, they visited and experienced activities in five local community assets to promote their engagement in these settings. An intervention guide specified all activities with its purposes and professionals and volunteers were specifically trained for their roles.

One intervention group was conducted in a semirural area (Cardedeu, zone A), and two in an urban area, Barcelona: one in a low socioeconomic level neighborhood (zone B) and one in a medium level one (zone C). Settings were selected by convenience to evaluate the viability of the intervention in different contexts.

### 2.3. Study Participants

The study population reached through the focus groups and interviews comprised 26 older people who participated in the program, nine older volunteers and six health and social care professionals. All 38 participants (37 women, 1 man) who started the program were involved in the participant-observation. Participants of the program were invited in person to the interviews and focus groups by the researcher (LCP) to take part in this qualitative study and agreed to participate. All volunteers and professionals directly involved in the program were invited to participate. The characteristics of all 38 participants have been previously described in detail. [30] Table 1 details the main characteristics of the 41 informants of the interviews and focus groups.

We intended to interview all 26 participants who finished the program out of 38 older people who started, but only 23 were available. None of the participants were excluded for any other reason. Moreover, one participant who had dropped out of each intervention group was selected taking into account their gender and the heterogeneous reasons for leaving the program: two women, one of whom dropped out to care for a family member and the other had an injurious fall, and one man who started a social activity. Furthermore, nine older volunteers who accompanied the three intervention groups were interviewed. One man and one woman initially involved as volunteers were not available. All six professionals involved as facilitators or observers were interviewed.

### 2.4. Data Collection Techniques

Focus groups and interviews were semistructured and followed a topic guide with open-end questions. The topic guide had been previously planned and prepared based on a review of the literature and the objectives of the study. The guide included adaptation according to the type of informant and had been agreed by the research team (Appendix A: “Topic guides of the semistructured interviews and focus groups with participants, volunteers and professionals”). The topic guide was pilot tested with the first two informants of each profile. Despite the structured script, the interviewer had the possibility to adapt the topics, add and change questions according to the progress of the group discussion. Focus groups with participants explored the perceived benefits on participants regarding loneliness, social support and participation, and health, accounting for contextual factors. In the interviews, participants were asked about their loneliness and participation prior to the program and the perceived benefits. Volunteers and professionals were asked about their perceptions of the process and benefits observed on participants.

Three focus groups with older participants and 41 semistructured interviews were conducted: 26 with older participants, six with professionals and nine with volunteers (one individual interview and three with small groups). Interviews and focus groups were conducted at the end of the intervention, in June–July 2012. Twenty-six older people were interviewed twice: in the focus groups conducted in their natural group during the last session of the program, and in an individual interview, in order to gain more personal information about their situation prior to the program, the process carried out and the perceived benefits.

Interviews with participants were partly conducted at participants’ homes and partly in a local senior club. Focus groups and interviews with professionals and volunteers were conducted in each primary health care center. Interviews lasted approximately one hour and focus groups around 1.5 h.

Moreover, participant-observation was conducted in all 15 sessions of the program in the three zones by one or two members of the research team. Field notes included any positive and negative information on the implementation of the intervention and the attitudes and reactions of participants, volunteers and professionals along the sessions. Notes were rigorously collected differentiating observed facts from subjective interpretations. A total of 58 field notes from observations were taken. All techniques were conducted by two female researchers (LCP, medical doctor, and GV, sociologist), both with experience in ageing research. As a consequence of the observation period, researchers established a rapport with participants during the 4.5 months. Participants were aware of the researchers’ involvement in the program.

### 2.5. Data Analysis

All conversational techniques were digitally recorded and transcribed (by DR, sociologist). A thematic content analysis was conducted. The analysis involved a triangulation of techniques, researchers, and informants. Two female researchers (DRA, sociologist, and LCP, medical doctor) independently analyzed the transcripts. DRA is an expert on qualitative research and was not involved in the program. The analysis was conducted according to following steps: (1) formulation of pre-analytical intuitions after successive readings of the transcriptions and the notes from documentary techniques; (2) creation of an initial analytical framework and text codification; (3) creation of categories by grouping the codes according to the analogy criterion based on predefined themes (experiences prior to the program, processes undergone during the intervention, influences of health, and context and perceived benefits according to the social capital theory) and new emerging elements from the discourses, with a continuous cross-checking between the categorization and the source of the data that combined a deductive and inductive approach; (4) analysis of each category and relationship with the others; (5) elaboration of the new text with the main results.

The loneliness model, the theories of behavior change, such as the social cognitive theory and the stages of change of the trans-theoretical model, and the three ageing crises were used to interpret the findings once data had been coded and categorized.

The results were structured to build an explanatory framework of the processes that participants underwent during the program, their perceived benefits and the main influencing factors. The results and the framework were discussed with the entire research team and verified with the corpus when needed. Informants verified results by providing their feedback on preliminary results. Informative richness for a deeper understanding of the phenomenon studied around the program was achieved according to the study aims and research questions. Data saturation was reached at the end of the analysis in the main categories for women, since participants were not contributing with new information.

### 2.6. Ethical Considerations

The ethics committees from IDIAP Jordi Gol and Universitat Autònoma de Barcelona approved the protocol (Ref number 1403). The informants participated voluntarily after signing informed consent forms. Anonymity, confidentiality and protection of stored data were guaranteed. No financial or material compensation was offered to informants.

## 3. Results

As recommended in the Standards for Reporting Qualitative Research [32] the results section comprises the synthesis and interpretation of the main findings with interpretations, inferences, and themes. Those are illustrated with quotes and field notes gathered in the qualitative procedures as evidence linking the results to the empirical data to substantiate the analytic findings and illustrate the process of interpretation based on these data. The verbatim quotations selected are the most representative of each theme, according to the richness of the idea or result they illustrate. Quotations from participants’ discussions included were translated by a professional scientific bilingual translator.

### 3.1. Participants’ Experiences of Participation and Loneliness prior to the Program

Two profiles of participants were identified regarding previous experiences of participation. The first profile was composed of participants with no previous experience of formal participation. They were women with a low educational level and mainly widowed. Their life had been focused on family and house care, and caring had been a barrier for participation. They shared trajectories of disempowerment, lack of courage to participate alone and renouncing to make decisions that they considered would be unfaithful towards others. Some women had no friends, had participated in social activities only with their husbands and stopped participating when they passed away. Some of them had no previous knowledge on community assets, or had prejudices, especially about senior clubs.


*“He didn’t want to go, because I sometimes said “let’s go and see”. We live beside the senior club… (…) but I didn’t have the strength to say “if you don’t come, then I’ll go on my own””. Participant 5, Woman, 78 years old, Zone C.*


The second profile had previous experience of social participation. They were mainly single, divorced or widowed, including the only widower. Widows who had participated together with their husbands in community assets had ended participation when their husbands passed away. Those who had participated on their own had conducted activities for other people (e.g., sewing), with others (e.g., social activities) or to help others (e.g., volunteering) and it had been a source of mental wellbeing. They had stopped mainly due to age-related health problems (e.g., chronic pain), economic problems, or translocation. Stopping them had contributed to their loneliness. Nevertheless, some participants reported having found ways of coping with limitations to maintain some informal activities, like overcoming pain to go for a walk.


*“For a long time I used to go there every day (to a center for disabled children) … look at my knee, I’ve needed an operation for 18 years but I decided not to have it, and I can’t feed them from sitting, because sometimes you have to hold their head and I can’t.” Participant 1, Woman, 83 years old, Zone C.*


Three main profiles of participants were identified regarding experiences of loneliness. In the first profile, participants expressed their loneliness as a consequence of widowhood. Their husbands’ absence had left a void that was impossible to fill and finding a new partner was disregarded to avoid being a “servant” again or because their husband was irreplaceable. They were living alone, suffered from loneliness mainly at home and coped with it by talking with their deceased husband, going out for a walk or having a pet.


*“I’m missing the most important thing, I’m missing my husband.” Participant 29, Woman, 78 years old, Zone B.*


In contrast, a recently widowed man who dropped out had joined the program to find a new partner.

Many of them had cared for family members and started to feel lonely after or while caring. They explained feeling lonely despite the support perceived and received from their family and neighbors. In some cases, widows suffered depressive symptoms and anxiety or had a pharmacologically treated depression. Nevertheless, it is to be mentioned that a minority of them expressed widowhood as a relief from a constrictive marriage.

The second profile comprised some long-term widowed, divorced or single participants who expressed that they were solitary. They expressed having a fear of relating with others, a lack of social relationships and that they received pressure from their family to interact more.


*“I’ve done it (joining the program) mainly because I had a problem relating with others, isn’t that right?” Participant, N. 18, Woman, 65 years old, Zone A.*


In the third profile, participants were suffering from loneliness in company. They had moved to live with their children due to health problems, or their children and grandchildren had moved to live with them due to economic problems. Older women expressed missing having their own space and a lack of communication with their children, who had little time for them.


*“My daughter and I have a good relationship, but I can’t have any conversations with her… She takes care of me if I am ill … but I can’t tell her stories about older people; they are very tedious, because she has no time. It’s true, she works long hours and has no time. She would like to listen to me and so on, but she says “Ah Mum, not today, I have no time, maybe on Sunday…”. “Participant 28, woman, 71 years old, Zone B.*


In addition, providing economic support to their children was a strong source of worry that intensified their loneliness.


*“And now I’m turning 74 years old. I thought than when I was old, I would have my retirement prepared, I thought I could live my life a bit. But I see it is the other way round, that now I have to be there for the others, instead of them being there for me; I am the one who has to be there for everyone.” Participant 2, Woman, 73 years old, Zone C.*


Table 2 summarizes the previous results regarding participants’ experiences when entering the program showing the identified profiles on loneliness and participation.

### 3.2. Perceived Benefits on Participants during and after the Program

Professionals and volunteers observed changes in participants that they attributed to the intervention. The benefits were more intensive among those participants who adhered more, suggesting a dose–response effect.

#### 3.2.1. Perceived Benefits on Social Support

Professionals and participants expressed that the program was especially successful at promoting mutual support. Living in the same area gave them a feeling of familiarity, and participants often met each other on the street, and sometimes walked back to their homes together.

According to participants, the group provided companionship, a feeling of social integration and sense of belonging to the group. The group was perceived as a space of attention, respect and affection to give and receive emotional support. When a participant suffered an injurious fall, was in low mood or had a new illness, support relationships could be observed.

Many participants were part of a group for the first time and for some participants, the group was the only place they had to socialize.

Participants discovered that peer relationships, as opposed to relationships within the family, provided a way of communicating shared worries and interests by sharing a similar age.


*“We are the same age, you can talk about the same things… youth, depending on the topic… you talk but…, I don’t know, youth is very different. (…) For me, the company of one or the other is different. With the group companions there …, I don’t know, maybe it’s another freedom, another thing because since we all speak about the same thing, pretty much, about what happens to us and about what we do not have…” Participant 29, Woman, 78 years old, Zone B.*


Participants identified others as a model to follow or, on the contrary, as a model to avoid, evoking positive changes.

Some participants became friends and started visiting and calling each other. While some people were previously aware of missing having friends, others made friends for the first time.


*“(...) because I don’t tend to go out with friends here and there. But now it’s different, since I’ve been coming here (…) Look, I get on very well with Maria, she’s a lovely and good woman and we get on great together. For her it’s the same; she says “I’ve found a shoe for my foot, because I don’t trust anybody but you”.” Participant 37, Woman, 77 years old, Zone B.*


In some cases, new friends generated subgroups that integrated other participants, including those who were more socially isolated. In other cases, friendships were closed, and some participants felt excluded.


*“... and they seem to have become very united to go out on walks together (…), but I go by and they are sitting there and never say “do you want to come with us”, so I go home....” Participant 2, Woman, 74 years old, Zone C.*


The group comprised different profiles regarding educational levels, age-related disability and health problems, which unified but also divided the group. Some participants expressed having felt united and treated without differences. In some cases, participants and volunteers developed support relationships with more vulnerable participants, moved by compassion. Telephone contact was especially relevant between participants with mobility limitations or living apart, and also for volunteers to support participants.


*“The one I see who needs to cheer up is Margalida, she is very down... (...) For me it’s no effort because it’s something I’ve done all my life, listen to people and be at their side and support them. Let them tell you things, especially that… I’ll go and see her this week, because she called me the other day and I went to her house and now I want her to come to my house”. Volunteer 2, Woman, 77 years old, Zone A.*


However, those participants with mobility limitations and hearing impairment were at higher risk of not establishing friendships and dropping out, thus losing the opportunity to benefit of the program at any level.

The few participants with a higher educational level expressed not sharing interests with the rest. For them, feeling valued and helpful for more vulnerable participants was key to remain in the program. In one group, there was a conflict with one participant. She felt more skillful and was jealous of those who participated more in the group.


*“You can see that she doesn’t stop talking, she always wants to speak… and from the first day there has been a conflict, and everybody saw there was a conflict. Even Jose said he didn’t feel comfortable because of her. And of course, this has restricted the dynamic a bit, hasn’t it? It hasn’t been easy…” Social care professional 1, Woman, Zone C.*


#### 3.2.2. Perceived Benefits on Loneliness

Most of the participants reported that their loneliness decreased after the program by feeling accompanied by peers and professionals, and thanks to the bonds established and to having become aware of and engaged in local activities of their interest. While some people said they no longer felt lonely because of new friendships, others continued to suffer from loneliness, but with less intensity. The awareness that loneliness was a common matter helped them to cope with it by realizing they were not alone in their loneliness.


*“I don’t feel lonely, now I have friends”. Participant 28, Woman, 71 years old, Zone B.*



*“Like bread and butter: loneliness is easier to digest when in company”. Participant 4, Woman, 78 years old, Zone C.*


Some participants expressed a transitory benefit on loneliness. For them, home was the space of loneliness, while the group and the street were relational spaces. Likewise, some participants said that the improvement would vanish once the group finished. Nevertheless, thinking and talking about the program with others also helped them to feel less lonely.


*“I am happy to join the group, but then, when I get back home, I fall apart, I need to be on the street with someone… at home, alone, is bad…” Participant 35, Woman, 81 years old, Zone B.*


Some widows who attributed loneliness to widowhood reported no decrease in loneliness after the program. Accordingly, in these cases the main effect desired of the intervention was not achieved. However, these participants reported other benefits such as an increase in social relationships, well-being and empowerment.


*“Since my loneliness is due to missing my husband, it cannot be replaced, at the moment, or ever.” Participant 13, Woman, 75 years old, Zone A.*


#### 3.2.3. Perceived Benefits on Social Participation

According to all types of informants, the program was generally successful at helping participants to discover and sometimes engage in local activities.

Visiting community assets allowed participants to get a sense of what was available and to remove prejudices. Moreover, some people returned to community resources where they used to go with their husbands.


*“The satisfaction of seeing things I had never seen before, although you imagine them, you’ve seen them on TV, but being there inside, you see it, you touch it, it is a big satisfaction…” Participant 5, Woman, 78 years old, Zone C.*


The visits included testing local activities and triggered participation in a wide range of activities. Some participants started participating in activities immediately and others started later during the program. They became engaged in activities that suited their interests, abilities or worries (e.g., memory training). Belonging to the group facilitated becoming engaged with other peers. Thus, new friends easily did new activities together, accompanying each other and reinforcing their friendship.


*“Carme and Teresa meet up to go to the cinema, since they live near each other, and Carme does not like going out on the street on her own at night. They meet up to see the film that the parish puts on in the cinema and has been recommended to them, but it’s not a planned activity; it’s an extra outing.” Field note, researcher LCP, referring to participants 10 and 13, Women, 75 and 80 years old, Zone A.*


Other participants made specific plans to start activities the following year and some exclusively connected with their wish to participate. For some participants, socializing was very important but participating in activities was not. Some participants, especially those who had been caregivers over the past years, discovered the value of doing activities with other people.


*“Everything we did there was new to me. Everything…” Participant 12, Woman, 79 years old, Zone A*


Low self-confidence and low communication ability, often related with low education, limited the benefits on participation to the extent of feeling they were not up to join community assets.


*“She tells me she’s odd and that she thinks everything is very nice and would like to get involved but she doesn’t feel capable because she is silly, she doesn’t express herself well, she talks poorly...” Field note, researcher LCP, referring to the participant 30, Woman, 84 years old, Zone B.*


#### 3.2.4. Perceived Benefits on Health

Participants, professionals, and volunteers agreed on the improvement in mental health. The program was seen as a strategy to prevent or alleviate depressive symptoms. Many participants took antidepressive drugs and/or tranquillizers and explained feeling better after the program. Some women expressed that the program was a salvation to them. One participant explained having solved her sleep problems.


*“For me, beforehand, I wasn’t able to go anywhere on my own. Now, I’ve changed! If I had to go for an X-ray, I had to be accompanied, and, since I have claustrophobia, in a lift and things like that… but now, I go alone wherever it may be, an X-ray, Sant Pau (Hospital)… I’m a different woman!” Participant 5, Woman, 78 years old, Zone C.*


According to the professionals, some participants were initially trapped in a loop linked to loneliness with an obsessive focus on illnesses and woes, but the intervention successfully broke it by connecting them with others, awakening the wish to remain connected and helping them to forget about their worries.

Sharing their woes and coping strategies among peers during the sessions was generally relieving and helped them to deal with them, although specific people needed to feel their suffering was greater.


*“By participating, you don’t feel lonely, with everything you are experiencing.” Participant 18, Woman, 65 years old, Zone A.*


Specifically, sharing the way in which they talked with their deceased husbands to overcome loneliness helped them to feel better instead of “crazy”, as they said.

In terms of positive mental health, participants reported an improved subjective well-being, becoming aware of worse circumstances and valuing their situation more. They reported being more understanding and empathic, and having more trust in other people; particularly those who were more closed and socially isolated. Others explained being more compassionate, respectful and having learned not to judge others. Likewise, they also reported feeling less worried and more able to deal with economic, family and health problems. Those living with family members expressed having learned to be more tolerant in cohabitation with other household members.

An empowerment process was observed that contributed to alleviating their loneliness. According to the three groups of informants, the program contributed to the development of personal potential and autonomy to participate and to live their life as they wanted, with less dependency on their children. They had a feeling of strength and of power to decide.


*“My daughter wanted me to spend every Sunday with them, but I didn’t like it and I used to say: “but why do I have to be here every Sunday?” and she’d say “so that you’re not on your own” (…) And now, if one day I don’t want to go for lunch I say “today, I won’t come for lunch, don’t wait for me because I’ll be with Maria”, now it’s different.” Participant 37, Woman, 77 years old, Zone B.*


Participants attributed their empowerment to the attention and value received. Additionally, realizing they had helped peers was very satisfying and increased their self-esteem, since it gave value to their life experience. Accordingly, feeling useful and able instead of useless meant that their life was not ending and was worth living.


*“(With the program) you have another stimulus, you feel like living, you feel like someone needs you for something. You feel that you, life, or God or whatever, needs you for something. Do you know what that feels like?” Participant 29, Woman, 78 years old, Zone C.*


In particular, those participants with a life trajectory that was family-oriented, said that they reached a new sense of freedom in their lives. Those participants with severe physical conditions felt connected with their wish to live by becoming aware that others do care about them. They were aware of their own empowerment process and participants mutually reinforced each other. It was strange for them having lived until then without these satisfying aspects of life. However, participants did not see themselves able to lead the continuity of the group and wanted someone as a leader to tell them where to go.

Empowerment was also enhanced by discovering new interests. Becoming engaged in local activities like physical activity and memory training especially promoted healthy ageing, but their physical activity also increased by starting to participate.

The program had some benefits on self-care and healthy lifestyles. Participants were motivated to dress smartly, some of them rediscovering the desire to get dressed up after widowhood by identifying some participants as a model to follow.

Two participants with hearing impairment felt motivated to wear the hearing aid that they had not used before because they wanted to feel connected to others in the group.

Through the program, they became aware of the relevance of taking care of their own health, especially those who had cared for a spouse and whose own health and self-care had not been a priority before.

Nevertheless, participants reported limited benefits on physical health, since many participants reported suffering from chronic conditions with aches that were difficult to alleviate.

Table 3 summarizes the results on the benefits of the program attributed to the intervention on social support, loneliness, participation and health. In each category, no effect, adverse effects, facilitators and mediators are specified when identified. Mediators are factors interpreted to be necessary in the pathway to reach benefits, while facilitators are factors considered as enhancing that area.

### 3.3. The Role of Urban, Semirural and Socioeconomic Context

Some differences and communalities could be identified between the three zones considering their semirural and urban contexts and the different socioeconomic levels in the urban neighborhoods.

As mentioned in Table 2, loneliness was worsened by a recent or prolonged translocation when the older person had not built a sufficiently fulfilling social life. This phenomenon was observed in all three zones, with translocations from urban to the semirural area, from rural areas to the city or when moving within the same city.


*“I say: so, you (meaning the husband who had died) were the one who wanted to live here (in the semirural area), you go, you leave me alone and I remain here”. Participant 13, Woman, 75 years old, Zone A.*


Only in the urban context, participants mentioned that the program contributed to a less hostile neighborhood. In the semirural area, many of the participants knew each other before, but the previous dynamics greatly influenced future relationships that could be built. Otherwise, in both urban areas, it was uncommon that participants knew each other and in these few cases, a previous relationship facilitated developing supportive friendships. Community assets were also already known in the semirural area but not their full range of offers. Whereas in the urban areas, many participants expressed surprise when discovering the opportunities for participation that they had near their homes.

Although the urban zones were different in terms of socioeconomic levels, observed processes and perceived impacts were more influenced by the socioeconomic characteristics of the participants than the contextual ones. Indeed, both zones were similar regarding some features that were relevant to enhance the effects of the program by promoting social interactions beyond the sessions. For instance, at that time, in both urban zones several community assets offered a diversity of activities and older people were present and relevant in the community life, e.g., participants easily met in their daily errands, sitting in benches or walking in pedestrian zones.

## 4. Discussion

### 4.1. Interpretation of Findings

The results of the qualitative evaluation of the program were convergent with the already published quantitative effects on loneliness, social support, and participation [30]. However, regarding reported health benefits at post-intervention, qualitative findings suggested changes that validated scales in the quantitative study could not detect at post-intervention. Nevertheless, at two years follow-up, the quantitative evaluation did detect a decrease in depressive symptoms in line with the qualitative findings. Accordingly, the main benefits of the program on mental health are in line with the protective effect of social capital on mental well-being among older adults [34].

Our results are consistent with research reflecting that handling loss is key in the attitude towards participation and social relationships [35]. Our study adds that interventions might encourage lonely people overwhelmed by loss to connect with meaningful activities and establish positive social relationships.

Our findings are consistent with the qualitative results of other programs in the same area like the Circle of Friends [28]. In both studies, participants felt alleviated sharing their diverse experiences of loneliness, although particular cases competed to be the worst case. Additionally, in both programs mutual support was observed, subgroups developed, and participants especially helped those who were more vulnerable. Meetings outside the groups were self-organized. Similarly, mild conflicts in relation with power games were rare but present and affected the group dynamic. Both studies showed that participants increasingly paid more attention to their appearance. Equally, the heterogeneity in age-related limitations influenced the group dynamics, limiting the participation of those more vulnerable participants.

The loneliness model could partly correspond to the type of loneliness observed by professionals prior to the program. The loneliness model proposes that chronic loneliness entails a cognitive bias consisting of a self-reinforcing loop associated with negative social expectations that cause social distance [36]. In this case, the self-reinforcing loop would be centered on illnesses and woes. However, participants were released from it at least during the program. Indeed, social relationships and participation seemed to create a positive self-reinforcing loop; opening participants up to others and to new experiences, relativizing their situations and encouraging them to get out of an introspective state, and thus involving more social relationships, and more participation that brought more meaning to their life. Accordingly, the observed empowerment process confirms the suitability of the empowerment model informing a successful design of the intervention.

The program helped participants to overcome, at least in part, the three ageing crises of autonomy, identity and belonging and consequently brought the feeling that life was worth living to participants and alleviated their loneliness [5]. It helped them to take care of their image and health, to take up their interests again, and provided them with the feeling of belonging to the group and their neighborhood. Mutual support helped them to overcome or cope better with their limitations and they felt more capable and useful.

The role of modeling, and the reported increased self-efficacy are in line with social cognitive theory [25]. Moreover, the stages of change of the trans-theoretical model supports the different levels of change described among participants: some participants started the action during the program (participation), others were in the preparation stage (were ready and made concrete plans), while others were in the contemplation stage (getting ready, connecting with their wish to participate) [26].

In line with the salutogenic approach, benefits were mainly reported on well-being, the social aspects of health and positive mental health, and there was also a decrease in ill mental health [4].

However, some participants did not report benefits from the program in certain spheres of their lives. Poor physical function and low socioeconomic level, especially when linked to low education, low self-confidence and low communication abilities, hindered engaging in the program and limited the process of change among participants. This is in line with previous research that shows that socioeconomic factors are key factors linking social relationships with health [4]. Nevertheless, it is to be remarked that this group of women had difficulties to access and continue formal education in their childhood and youth, which explains their low educational level. However, those with certain personal capabilities and social abilities could further develop and grow with the program. Moreover, some widowed participants who attributed loneliness to their widowhood continuously felt lonely although expressing an increase in social relationships. The distinction between social and emotional loneliness could partly explain why these cases remained emotionally but not socially lonely. While social loneliness occurs when the number of relationships with family, friends and colleagues is smaller than desired, emotional loneliness refers to situations where the wished intimacy in confidant relationships is not realized [37].

Lastly, the historical and cultural context seems to configure a generation of older women who had grown up assuming traditional roles of dependence on their husbands. Some of them remained powerless in widowhood, while others were relieved, and others managed widowhood well alone over time. In addition, the 2008–2009 financial crisis seems to have worsened the experience of the ageing process and enhanced loneliness by stressing family dynamics.

### 4.2. Strengths and Limitations

The rapport built between researchers and participants during the program generated a trust that facilitated the observation of the sessions and sharing personal experiences in the interviews, although it might also have influenced their answers, consciously or unconsciously wanting to please researchers. Nevertheless, the assumptions we had as researchers regarding how and why the program should have reduced their loneliness were challenged from the first group session to the last interview.

Among informants, men were rare, since women were a clear majority among participants and the only gender among volunteers, professionals and researchers. This fact has as consequences that the men’s discourse is underrepresented in the results. However, community-based programs targeting older people in Spain are frequently dominated by women [38]. Accordingly, our study contributes to the understanding of the experiences of women, who are the majority of users of this type of program. Moreover, older people who adhered to the intervention were the majority among informants. Nevertheless, three people who dropped out for different reasons were interviewed, and the participant observation technique involved all participants since all sessions were observed.

Benefits reported by older people at the end of the program were triangulated with those perceived by volunteers and professionals and with the observations of researchers during the process. Accordingly, the richness and complementarity of the information generated with the different techniques and the three types of informants are a strength of the study, since triangulation of informants and techniques is a criteria for rigor and quality in qualitative research to enhance trustworthiness and credibility of data analysis [31]. Nevertheless, the constructivist research paradigm framing qualitative research does not aim to study a representative sample and generalize the results, but to study in depth a phenomenon in a given context that can be transferred to similar contexts [39]. Moreover, qualitative results can guide further quantitative research to objectify, quantify and generalize the magnitude of effects.

Qualitative findings are limited to the post-intervention timepoint with no further data on whether and how the perceived effects lasted and what was the trajectory of participants not experiencing certain effects. However, the quantitative evaluation was repeated at two years follow-up and significant long-term effects on loneliness, social participation and depressive symptoms were detected [30]. Moreover, almost half of the participants maintained long-term contact with at least one person from the group and 40% continued participating in activities [30].

Lastly, primary care professionals involved in the program were especially motivated to work on loneliness, and the implementation of the program might face barriers in primary health care contexts with a strong biomedical focus. Accordingly, caution is required before transferring these results to other settings. Nevertheless, similar community-based programs in different contexts, such as Circle of Friends, have shown their applicability.

### 4.3. Implications for Research, Practice and Policy

This program supports the WHO Active and Healthy Ageing policies and provides insight into how to enhance social networks and participation while ageing to enhance well-being.

In addition, our findings should support current practices and policies of social prescribing programs, which link primary care patients with community resources with the aim of strengthening participation and social support, and promoting health, particularly mental health, and well-being [40].

Nevertheless, the role of primary health care in loneliness interventions may differ according to the cultural context and the characteristics of the health and social care system and the available community resources [41]. In any case, attention must be placed on not medicalizing loneliness when interventions are developed in primary health care.

Regarding the intervention design, guaranteeing the continuity of the group remains a challenge, as well as an appropriate follow-up to enhance, if needed, participants’ engagement in the social activities in community assets. Strategies are needed to focus on those persons with social and health vulnerabilities and, consequently, at risk of dropping out or of being socially excluded during or after the program.

### 4.4. Future Research Directions

Future research should include more qualitative evaluations of interventions for a better understanding of personal processes and perceived intervention benefits on loneliness and unintended effects, addressing its complexity, including context specificities [42].

Programs addressing loneliness tend to reach and work well with certain profiles of people (e.g., widow women), while they might be missing some others [43]. Indeed, loneliness is crossed by inequality axes such as gender, age, social class, disability and ethnicity [44]. Some of these profiles might be “hard-to-reach” if they are not specifically targeted by programs. In this vein, a deeper understanding of the perspective of men and participants dropping out of programs tackling loneliness is fundamental to rethink interventions to reach them, identify profiles at risk of dropping out and address their potential reasons. Therefore, there is an urgent need to move towards personalizing interventions with and for older people with an equity perspective. Moreover, future studies should assess the long-term effects of such interventions, also from a qualitative perspective, especially when the ageing process might further affect their abilities and opportunities to participate and socialize.

Finally, with the expectation of an increasing number of vulnerable older people vaccinated against COVID-19 in Europe, research should guide post-pandemic times on how to rebuild social capital, especially in older people with aggravated loneliness during the pandemic.

## 5. Conclusions

The qualitative evaluation of the Paths program has contributed to understanding the complex processes that are involved when promoting social capital in older people with low participation and loneliness. The intervention tried to promote social capital to make it a social resource available to all group members. Different degrees of success were observed among participants on their reported alleviation of loneliness, increase in social relationships and engagement in social activities. In the most successful cases, the program enabled their empowerment and enhanced processes of change. Those participants reported an improvement in mental well-being, experienced new freedoms and became reconnected with the sense that life was worth living. However, some widowed participants remained emotionally lonely and other participants were not interested in joining social activities. Moreover, vulnerabilities related to health, socioeconomic factors and age-related disability limited the adherence to the program and the perceived benefits of the intervention.

These findings should support further designs, and the implementation and evaluation of interventions. The cultural context of the study is a familistic society with a primary health and social care system with a community-based approach in Catalonia, Spain. However, our results can inspire other programs that should be flexible to adapt intervention components to the specific contexts and to participants’ characteristics.

Our results might guide post-pandemic programs on how to resume face-to-face social interactions and social activities among older people suffering from loneliness.

## Figures and Tables

**Table 1 ijerph-18-05580-t001:** Characteristics of participants, volunteers, and professionals interviewed.

Context	Technique	Number of Informants	Age	Gender	Educational Level/Occupation **
Zone A: Semirural context with a medium socioeconomic level.	Participants *				
	One focus group	Five participants	65–74 y.: 1	Five women	One with medium education and four with low education
			75–80 y.: 2		
			over 80 y.: 2		
	Eight individual semistructured interviews	Eight participants	65–74 y.: 1	Eight women	One with medium education and seven with low education
			75–80 y.: 5		
			over 80 y.: 2		
	Volunteers				
	One interview in small group	Four volunteers	65–74 y.: 1	Four women	Low education
			75–80 y.: 2		
			over 80 y.: 1		
	Professionals				
	Two individual semistructured interviews	Two professionals from primary health care and social services	30–50 y.: 1	Two women	One nurse
			51–65 y.: 1		One social worker
Zone B: Urban context with a low socioeconomic level.	Participants *				
	Focus groups	Nine participants	65–74 y.: 2	Nine women	Low education
			75–80 y.: 4		
			over 80 y.: 3		
	Individual semistructured interviews	Eleven participants	65–74 y.: 2	Eleven women	Low education
			75–80 y.: 6		
			over 80 y.: 3		
	Volunteers				
	One interview in small group	Two volunteers	63 and 80 years old	Two women	Medium and low education
	Individual semistructured interview	One volunteer	63 years old	One woman	High education
	Professionals				
	Two individual semistructured interviews	Two professionals from primary health care	30–50 y.: 1	Two women	Two social workers
			51–65 y.: 1		
Zone C: Urban context with medium socioeconomic level.	Participants *				
	One focus group	Seven participants	65–74 y.: 1	Six women and one man	One with high education, six with low education
	Seven individual semistructured interviews		75–80 y.: 2		
			over 80 y.: 4		
	Volunteers				
	One interview in small group	Two volunteers	65–74 y.: 1	Two women	Medium education
			75–80 y.: 1		
	Professionals				
	Two individual semistructured interviews	Two professionals from primary health care	30–50 y.: 2	Two women	One social worker and one nurse
			51–65 y.: 0		

* Note: All participants who were individually interviewed had previously participated in the focus groups, except three from zone A and two from zone B, who were only individually interviewed. ** “Educational level” applies to older participants and volunteers and “occupation” refers to professionals.

**Table 2 ijerph-18-05580-t002:** Summary of results regarding participants’ experiences prior to the program.

Participants’ Experiences Prior to the Program		
Experiences of participation	No previous experience of formal participation	Knowledge about local community assets: no knowledge, perceived barriers or prejudices
		Life focused on family and house care
	Previous experience of social participation but stopped	Participation linked to husband (stopped when widowhood)
		Due to health-related limitations
		Due to changing neighborhood
		Due to economic constraints
Experiences of loneliness	Loneliness attributed to widowhood	
	Participants who expressed that they were solitary but wishing more social relationships	
	Suffering from loneliness in company	Lack of communication
		Lack of own space
	Factors worsening loneliness	Economic constraints, e.g., providing economic support to family
		Urban–rural translocation with insufficiently built social network

**Table 3 ijerph-18-05580-t003:** Summary of results regarding perceived benefits of the program.

Perceived Benefits of the Program	
Perceived benefits on social support	Company
	Social integration
	Sense of belonging
	Support relationships: ○ Friendship: participants with affinity becoming friends, including or excluding others *(adverse effects)* ○ Compassion: Relationship with more vulnerable participants moved by compassion
	Conflicts *(adverse effects)*
Mediator	Social network among peers from the same neighborhood
Facilitators	Previous knowledge among participants
Perceived benefits on loneliness	Loneliness decreased
	Transitory improvement in loneliness: during the program or during the group sessions
	No improvement in loneliness (in case of loneliness attributed to widowhood)
Perceived benefits on participation	Do not want to participate *(no effect on participation)*
	Connecting with the wish to participate
	Plans for participating
	Started participation
Mediator	Knowledge on local community assets
Facilitator	Local activities that meet interests, abilities and worries
Perceived benefits on health	Disconnect from worries and discomfort
	Self-reported improvement of mood and decrease depressive symptoms
	Better strategies to affront health and personal problems
	Increase trust in others
	Better self-care and healthier lifestyles
	Feeling useful, able and strong; life is not ending, life is worth living
Mediator	Empowerment process, autonomy to participate, feeling of strength and of the power to decide
Barriers	Vulnerabilities: Age-related health limitations: acoustic limitations, chronic diseases and mobility disability Low education: poor communication ability Personal resources: low self-efficacy and poor coping strategies

## Data Availability

The data presented in this study are available on request from the corresponding author.

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
