# Peer review of "“Not Alone in Loneliness”: A Qualitative Evaluation of a Program Promoting Social Capital among Lonely Older People in Primary Health Care"

_ijerph, 2021, doi:10.3390/ijerph18115580_

Round 1

Reviewer 1 Report

In my opinion, authors addressed all the concerns raised in the previous version.

Author Response

Thank you very much. We are very grateful for your comments, which have undoubtedly contributed to improve the manuscript.

Reviewer 2 Report

The subject of loneliness is topical and important and it is great to see programmes being put in place to address this area.  Details of what the programme actually entailed, as related to the theories outlined should be included - were particular activities designed for specific purposes?  In addition the mapping of theories to methods and results should be more explicit e.g. how specifically do the findings relate to the three crises mentioned in the interaction and the discussion and how stage of change assessed. Significant editing for English phrasing and grammar is required.

Author Response

Comments and Suggestions for Rewier 2

The subject of loneliness is topical and important and it is great to see programmes being put in place to address this area.  

ANSWER:

Thank you for your positive comment.

Details of what the programme actually entailed, as related to the theories outlined should be included - were particular activities designed for specific purposes?  

ANSWER:

We have revised the description of the programme and added some aspects in this section. This current version includes all key aspects of the intervention according to the checklist of the reporting template for intervention description and replication guideline (TIDieR https://www.equator-network.org/reporting-guidelines/tidier/ ) from the Equator Network. Line: 255-272.

In addition the mapping of theories to methods and results should be more explicit e.g. how specifically do the findings relate to the three crises mentioned in the interaction and the discussion and how stage of change assessed.

ANSWER:

We have done our best to address this comment. We have added in methods which framework guided the analysis (design section, Line: 254) and how other theories such as the ones on behavior change have been incorporated in the analysis (analysis section Line: 408-410).

Significant editing for English phrasing and grammar is required.

ANSWER:

An external person with proficiency in English has conducted the proof reading of the manuscript.

Reviewer 3 Report

This paper presents a qualitative study of the effectiveness of an intervention programme targeted at older adults who are experiencing loneliness in Spain. The results showed that the programme had varying success across different older adults with differing social profiles, but generally is helpful in decreasing loneliness. Overall, the paper was well-written and provided insights on loneliness alleviation in older adults with varying experiences.

The authors have done well to present loneliness and its significance in the introduction in a concise manner. The authors may like to consider merging shorter paragraphs together to increase readability.

A few minor comments about the introduction:

  • In lines 54 – 57, it would be good if the authors rephrased to make the point clearer; it is unclear how increasing social expectations and active social participation are both vital for the ageing experience.
  • The term “social capital” is mentioned a few times in the introduction. As it may carry different meanings based on the source of reference, it would be good to properly define the term in accordance with its role in this paper.
  • Given that the qualitative study also investigated the role of demographic characteristics (e.g., living arrangement, marital status, education, pre-intervention social network, etc.) on the effectiveness of the intervention, it would be a waste to not mention this effort in the introduction.

The materials and methods section was clear and well elaborated on. The authors should take note that “rigour” (noun) in line 99 should be “rigorous” (adjective), when used to describe the criteria used for the study.

The results section was engaging and highlighted the differences in experience in older adults with varying profiles. The authors may wish to consider greater comparisons by the zones (i.e., semi-rural with urban and low with medium socioeconomic level) in which the programme was executed. Given a relatively even distribution of participants across the three zones, it might be a worthy comparison to have that would bring about interesting insights about socioeconomic differences in intervention effectiveness. Additionally, the initial investigation into the participant’s experiences prior to the programme could be brought back when discussing the benefits of the programme. For instance, comparisons between the older adults with differential experiences could provide illuminating insights.

Similarly, in the discussion section, more could be done to elaborate on the differences between participants who benefited more and those who benefited less or not at all from the programme.

The strengths and limitation of the study was well recognised. The authors might want to mention that a separate limitation of the study is that the effects of the intervention tested were immediate (within a short timeframe post-intervention), no long-lasting effects were tested in the study. As such, based on the Loneliness Model as proposed by the authors, there could be a high chance that the participants might relapse into loneliness. Future studies could be suggested to study the longer-term effects of such interventions, especially due to older adults’ reduced ability to participate in such programmes as they age.

Additionally, the paper would benefit from a more thorough proof-reading as different forms of spelling could be seen throughout the paper, such as rigour being spelt “rigour” in line 99, but “rigor” in line 594.

Reviewer 4 Report

I find this to be an interesting and well presented report and have no objections, corrections or other comments to make. Although my experience in qualitative research is very limited, I don't hesitate to recommend this work for publication in the journal.

Author Response

Thank you for your positive comments on the manuscript.

Round 2

Reviewer 3 Report

The authors have addressed most of my concerns and comments in the revised manuscript. Well done!

This manuscript is a resubmission of an earlier submission. The following is a list of the peer review reports and author responses from that submission.

Round 1

Reviewer 1 Report

Given the qualitative nature of the study it should be better to avoid the use of casual terms such as "effect" or "influence". It this sense, aims of the study should be reviewed.

The program was developed in 2011-2012, nine years ago. Please, explain why the qualitative evaluation is now reported and discuss about how findings could differ nowdays.

Please, explain how the gender bias in the participants composition could affect the results.

Please, explain how the selection of the professionals and voluteers was made.

Please, explain the type of work developed by professionals and volunteers in the programa ejecution.

Please, explain if any observation sheet was followed for observations, the type of information registered and if "observators" were part of the research team or not, an how this could affect the results of the current evaluation.

Please, clarify if the focus group interview was semi-structured or fully structured following only the script included in the appendix, and what others questions were elucidated during the focus group.

Please, specify if the participants obtained any kind of compensation to participate in the study.

In relation to the focus groups; did the data collection ended when participants did not longer reveals new patterns, themes or other findings?

Please, include the interview script in an appendix and explain how the interview script was developed.

Results:

Figure 1 is difficult to read. Authors should consider to split the figure in two to improve readiness. More description of the figure is needed in the first paragraph of the results.

Please, explain how verbatim quotation were selected to ilustrate the resuts.

I should temper the discussion in terms of casualty. the language used is more appropiate for a quantitative study than a descriptive-qualitative. Without longitudinal data authors should temper the description in terms of "impact".

Conclusions should be further developed discussing cultural and contextual factors that could affect results, what it is really important for a qualitative study.

Finally, the conclusion could be improve. What is the take-home message of this paper? Here, some conclusive remarks may be specified by the authors for the general readership.

I wish you all the luck revising your paper.

Reviewer 2 Report

The manuscript entitled “<<Not alone in loneliness>>: a qualitative evaluation of a programme promoting social capital among lonely older people in primary health care” presents interesting issue, but some areas must be corrected.

Major:

  1. Based on the presented information, it seems that Authors present the results of a program which was conducted in a group of 38 participants and which was finished by only 26 participants. Taking this into account, we can not conclude about it and formulate any general observations, as it is not possible based on such a small group of respondents.
  2. It seems that taking into account small number of potential respondents and the fact that only 23 agreed to participate in the study, Authors included also health and social care professionals, and volunteers, causing that the studied group is not representative at all.
  3. Authors did not verify representativeness of their studied group at all, so we can not state that this group represents any typical characteristics.
  4. Authors planned some focus groups with their participants, but based on such small studied group, not representative and mixed with other respondents (health and social care professionals, and volunteers), we can not conclude.
  5. Authors presented a highly subjective observations, as their gathered group and applied research techniques do not allow to conclude about. Authors should be aware, that if they want to have focus groups, participants of those groups must be selected based on very specific criteria, and we can not include 23 respondents, as we had only 23 respondents available.

Title:

Authors should formulate more “scientific” title - formulated while using a proper scientific language, as their current title is rather formulated as for the column of the newspaper. The proper title should be rather informative than catchy. “Not alone in loneliness” does not present any information in this title.

Abstract:

Authors should provide more information about the program

It should be clearly indicated what was the number of participant’s if there were 41 persons, but only 36 semi-structured interviews – what about other 5 participants?

“persons were included comprising older people, health and social care professionals, and volunteers” – number of participant’s in sub-groups should be presented

Authors should present more specific details associated with their research techniques.

Authors should provide any numeric results to indicate e.g. share of respondents indicating any observations.

Authors should provide more reliable conclusions that may be formulated based on their research material.

Introduction:

Authors should reduce the number of information that are presented in Introduction Section – they should present only essential information to justify their study

Authors failed to justify the need for their study – they should present what is already known and what are the “gaps” in the scientific knowledge to formulate the aim of their study.

Lines 86-101 – Authors should not present their program in this section, but rather in Materials and Methods Section

Lines 102-104 – does not present any specific information – should be removed

Materials and Methods:

There are the serious doubts associated with the materials and methods of the presented study (see above) – Authors should address them in this section and present related issues clearly

Authors should present all necessary details associated with the programme – it should be presented in such way to be able to be reproduced by other researchers

Authors should refer specific number of ethical committee agreement

Results:

Figure 1 – this figure is hard to follow, to understand and to understand what is the role of this figure within the study (what is presented here)

The whole Results Section seem to present rather the own opinions of Authors than any real results of the study

There are the serious doubts associated with the presented study (see above) – Authors should address them in this section and present related issues clearly

Discussion:

Authors should not reproduce results in this section

Authors should present more balanced opinions and do not overestimate the results of their own study

Authors should in their discussion include 3 areas: (1) compare gathered data with the results by other authors, (2) formulate implications of the results of their study and studies by other authors, (3) formulate the future areas which should be studied

Authors should clearly discuss the limitations of their study (as there are really major limitations – this section should be broaden), while including abovementioned issues

Conclusions:

Authors should present more balanced opinions and do not overestimate the results of their own study

Authors contributions:

It seems that contribution of some Authors was only minor (DRA, RM) and they did not participate in preparing manuscript. There is a serious risk of a guest authorship procedure which is forbidden. In such case they should be rather presented in Acknowledgements Section and not be indicated as authors of the study.

Who is XX?

Round 2

Reviewer 1 Report

Authors have addressed all my concerns. I have not further comments.

Author Response

We are very glad to read that the first reviewer was satisfied with the doubtless improvements derived from his/her comments.

Reviewer 2 Report

The manuscript entitled “<<Not alone in loneliness>>: a qualitative evaluation of a programme promoting social capital among lonely older people in primary health care” presents interesting issue, but some areas must be corrected. Unfortunately Authors ignored my previous comments.

Major:

  1. Based on the presented information, it seems that Authors present the results of a program which was conducted in a group of 38 participants and which was finished by only 26 participants. Taking this into account, we can not conclude about it and formulate any general observations, as it is not possible based on such a small group of respondents.
  2. It seems that taking into account small number of potential respondents and the fact that only 23 agreed to participate in the study, Authors included also health and social care professionals, and volunteers, causing that the studied group is not representative at all.
  3. Authors did not verify representativeness of their studied group at all, so we can not state that this group represents any typical characteristics.
  4. Authors planned some focus groups with their participants, but based on such small studied group, not representative and mixed with other respondents (health and social care professionals, and volunteers), we can not conclude.
  5. Authors presented a highly subjective observations, as their gathered group and applied research techniques do not allow to conclude about. Authors should be aware, that if they want to have focus groups, participants of those groups must be selected based on very specific criteria, and we can not include 23 respondents, as we had only 23 respondents available.

Title:

Authors should formulate more “scientific” title - formulated while using a proper scientific language, as their current title is rather formulated as for the column of the newspaper. The proper title should be rather informative than catchy. “Not alone in loneliness” does not present any information in this title.

Abstract:

Authors should provide more information about the program

It should be clearly indicated what was the number of participants if there were “36 interviews applying a semi-structured topic guide involving 26 participants (older people), six health and social care professionals and nine volunteers” (as 36 is less than 26+6+9)

Authors should present more specific details associated with their research techniques.

Authors should provide any numeric results to indicate e.g. share of respondents indicating any observations.

Authors should provide more reliable conclusions that may be formulated based on their research material (e.g. they can not conclude about Spain, but only about their programme)

Introduction:

Authors should reduce the number of information that are presented in Introduction Section – they should present only essential information to justify their study

Authors failed to justify the need for their study – they should present what is already known and what are the “gaps” in the scientific knowledge to formulate the aim of their study.

Authors should not present their program in this section, but rather in Materials and Methods Section

Materials and Methods:

There are the serious doubts associated with the materials and methods of the presented study (see above) – Authors should address them in this section and present related issues clearly

Authors should present all necessary details associated with the programme – it should be presented in such way to be able to be reproduced by other researchers

Results:

Figure 1 – this figure is hard to follow, to understand and to understand what is the role of this figure within the study (what is presented here)

The whole Results Section seem to present rather the own opinions of Authors than any real results of the study

There are the serious doubts associated with the presented study (see above) – Authors should address them in this section and present related issues clearly

Discussion:

Authors should present more balanced opinions and do not overestimate the results of their own study

Conclusions:

Authors should present more balanced opinions and do not overestimate the results of their own study
